# Mechanosensitivity and Adipose Thickness as Determinants of Pressure to Reach Deep Fasciae in Cervical and Thoracolumbar Regions

**DOI:** 10.3390/s25165073

**Published:** 2025-08-15

**Authors:** Sílvia Pires, Marta Freitas, Francisco Pinho, Sofia Brandão

**Affiliations:** 1Escola Superior de Saúde do Vale do Ave, Cooperativa de Ensino Superior Politécnico e Universitário, Rua José António Vidal, 81, 4760-409 Vila Nova de Famalicão, Portugal; marta.goncalves@ipsn.cespu.pt (M.F.); francisco.pinho@ipsn.cespu.pt (F.P.); 2H2M—Health and Human Movement Unit, Polytechnic University of Health, Cooperativa de Ensino Superior Politécnico e Universitário, Cooperativa de Responsabilidade Limitada, 4760-409 Vila Nova de Famalicão, Portugal; 3Centre of Research Rehabilitation (CIR), Escola Superior de Saúde, Rua Dr. António Bernardino de Almeida 400, 4200-072 Porto, Portugal

**Keywords:** deep fasciae, pressure pain threshold, mechanical pressure, adipose tissue thickness

## Abstract

Objective: We aimed to quantify the mechanical pressure required to reach the deep cervical and thoracolumbar fasciae, to examine its association with pressure pain threshold (PPT) and adipose tissue thickness, and to determine whether PPT and adipose tissue thickness can predict the mechanical pressure needed to reach the fascia. Methods: Forty-three subjects’ PPT, mechanical pressure and skinfold in the trapezius and lumbar regions were evaluated using an algometer, an ultrasound scanner, and an adipometer. The Wilcoxon test, Student’s *t*-test, and Pearson and Spearman’s correlation tests were used (*p* < 0.05). Results: The values of mechanical pressure in the cervical and lumbar regions were 6.06 ± 0.186 N/cm^2^ and 5.85 ± 5.280, 7.287 N/cm^2^, respectively. PPT values were 18.88 ± 0.917 N/cm^2^, and 46.46 ± 2.408 N/cm^2^, respectively (*p* < 0.001), and the adipose tissue thickness values in the cervical and lumbar regions were 0.88 ± 0.675 cm, 1.08 and 1.48 ± 0.069 cm (*p* < 0.001). There was no correlation between the variables in either region under study. Conclusions: The mechanical pressure required to reach the deep cervical and thoracolumbar fasciae was similar with an average value of approximately 6 N/cm^2^, suggesting a consistent mechanical response across these anatomical regions. Mechanosensitivity and subcutaneous adipose tissue thickness did not influence the mechanical pressure needed to access the deep fascial tissue.

## 1. Introduction

Fascia is a highly viscoelastic, uninterrupted connective tissue, forming a dense three-dimensional network of collagen, which plays a crucial role in the organisation, support, protection and movement of different systems [1,2,3]. In addition to its mechanical and structural functions, fascia is richly vascularised and innervated, containing blood vessels and sensory nerve endings - including mechanoreceptors and nociceptors—that contribute to its role in proprioception and pain modulation [4]. 

Several authors argue that, in addition to its structural role, the fascia also functions as an active sensory organ. This is due to the presence of mechanoreceptors, proprioceptors, nociceptors and interoceptors, which have a key role in pain perception, neuromuscular modulation and body awareness [2,5,6,7]. Recent evidence proposes a view of fascia as a body-wide layered system of connective tissues that enables tensional transmission, shearing mobility, and interstitial fluid flow across multiple scales [8]. This systemic perspective includes superficial, deep, visceral and neural components, as well as the fascial *interstitium*, hyaluronic acid-rich matrix involved in hydration, mechanotransduction and tissue communication [8].

The deep cervical and thoracolumbar fasciae are especially important due to their biomechanical role in spinal stability and movement. The deep cervical fascia provides structural support and facilitates cervical mobility, while the thoracolumbar fascia plays a crucial role in transmitting and distributing mechanical forces between the trunk and the lower limbs [6,9,10]. Nociceptive input does not arise exclusively from the fascia but from multiple tissue layers, including the epidermis, dermis, subcutaneous tissue and deeper structures. These layers differ in embryological origin and innervation: the epidermis, derived from the ectoderm (like the central nervous system), is innervated by autonomic Aδ and C fibres responsible for nociception and thermal stimuli. In contrast, the dermis and subcutaneous tissues, of mesodermal origin, are innervated by somatic Aβ fibres that respond to mechanical stimuli. This layered neuroanatomical arrangement explains the varied sensory responses elicited by manual pressure and highlights that any attempt to reach the deep fascia inevitably stimulates multiple nociceptive sources. As such, isolating fascial contribution in vivo remains methodologically challenging [11,12].

According to the literature, any restriction of the cervical and thoracolumbar fasciae, such as adhesions, stiffness or inflammation, can lead to mechanical hypersensitivity, which is a determining factor for the presence of pain, increased muscle tension, decreased joint mobility, flexibility deficits and altered nociception [5,13,14,15].

Among the various therapeutic modalities for myofascial treatment, application of manual pressure to the fascia is widely used in myofascial release techniques, with the aim of altering the density, viscosity, sensitivity and mobility of the tissue, as well as modulating the perception of pain [16,17]. However, it is important to note that any manual pressure applied to reach the fascia must first pass through the overlying skin layers: the epidermis, dermis, and subcutaneous tissue. These structures are not merely passive barriers but actively contribute to the autonomic response—initially via the epidermis and subsequently through mechanosensory input from the dermis. During myofascial therapy, the practitioner’s hand is in direct contact with the epidermis. While deeper structures such as the dermis, subcutaneous tissue, and fascia are the intended therapeutic targets, they are indirectly affected by the mechanical transmission of pressure [12].

The presence of subcutaneous fat above the superficial fascia also seems to play a significant role in mechanosensitivity [18]. Studies suggest that changes in the amount and distribution of subcutaneous fat can alter the tension on the fascia and may interfere with mechanosensitivity and restrict fascial mobility, thereby reducing the ability to adapt to movement [19,20].

Although the role of fascia in pain modulation, particularly its mechanosensitivity and relationship with adiposity, has been increasingly explored in the literature, to our knowledge, few studies have directly quantified the mechanical pressure required to access the cervical and thoracolumbar deep fasciae. Existing research shows that fascial manipulation can reduce pain and modify fascial stiffness and thickness, particularly in individuals with chronic neck or low back pain [1,15]. However, studies are lacking that isolate the central autonomic effects of the epidermis on pain and mobility [21], and subsequently correlate the mechanical pressure required to reach deeper fascial layers with tissue thickness and subjective pain perception. This knowledge could facilitate the development of evidence-based guidelines for more effective manual therapy interventions for neck and low back pain [8,22,23].

Therefore, this study aimed to quantify the mechanical pressure required to reach the deep thoracolumbar and cervical fasciae, to examine its association with mechanosensitivity and adipose tissue thickness, and to determine whether mechanosensitivity and adipose tissue thickness can predict the mechanical pressure needed to reach the fascia, using linear regression analysis.

## 2. Materials and Methods

### 2.1. Study Design and Ethics

This cross-sectional observational study was conducted at the Escola Superior de Saúde do Vale do Ave (ESSVA), part of the Polytechnic Institute of Health of the North (IPSN), Portugal. Data were collected between October and December 2023. Forty-three participants were assessed in a single experimental session, all of which were conducted at approximately the same time of day (morning) to minimise potential circadian effects on fascial tone [24]. The study was designed and reported in accordance with the Strengthening the Reporting of Observational Studies in Epidemiology (STROBE) guidelines (BIBLO).

The study was approved by the Ethics Committee of the IPSN (references: 41 and 48/CE-IPSN/2023). All participants provided written informed consent prior to participation, in accordance with the Declaration of Helsinki.

### 2.2. Participants

Participants were volunteers recruited through convenience sampling by direct invitation, using academic contacts. A total of forty-three university students of both sexes were available to participate. Inclusion criteria were being a healthy adult aged between 18 and 30 years. Exclusion criteria included the presence of neurological disorders, neck, shoulder or lumbar pain at the time of assessment, torticollis, inflammatory rheumatic conditions, history of spinal surgery, and fibromyalgia [25]. No formal sample size calculation was performed, as this was an exploratory observational study. The sample size was determined pragmatically, based on the feasibility and accessibility of the population. This approach is considered appropriate in early-phase exploratory research, where the aim is to generate preliminary insights and guide future hypothesis-driven studies. Moreover, the use of healthy university students ensured homogeneity in terms of age and activity levels, reducing potential confounding variability.

### 2.3. Instruments

#### Sample Selection and Characterisation

Sample characterisation was carried out through a structured questionnaire that included sociodemographic and anthropometric data, smoking habits, lifestyle, general health status, and exclusion criteria. This non-validated instrument was used solely for descriptive purposes to characterise the sample and ensure eligibility. No inferential analyses were conducted based on these data.

To assess the main outcome variables, we used a digital algometer, an ultrasound scanner, and a skinfold calliper (adipometer).

To access the pressure pain threshold (PPT), the digital algometer was a Wagner FORCE ONE FDIX (Wagner Instruments, Greenwich, CT, USA) equipped with a round rubber tip of 1 cm^2^.

In the second stage, ultrasound imaging was performed using a General Electric VIVID E scanner (GE HealthCare, Chicago, IL, USA) with a linear probe (5 cm^2^). The probe was adapted to support the algometer, allowing simultaneous monitoring of fascial deformation and quantification of the mechanical applied pressure (Figure 1). To accurately quantify the pressure applied to the fasciae, the force displayed by the algometer (in kgf) was divided by the fixed contact area of the ultrasound transducer (5 cm^2^), yielding pressure in N/cm^2^. This setup enabled synchronised measurement of both force and fascial deformation in real time. The system was pre-tested to ensure mechanical stability and reproducibility, and the adaptation ensured that the pressure was applied perpendicularly to the tissue without compromising image quality or measurement reliability.

A gradual increase in pressure was applied perpendicularly until fascial contact was confirmed sonographically, defined by loss of fascial sliding or visible deformation of the fascial plane. This criterion was standardised during a pilot phase and applied consistently by a single experienced radiologist. Pressure was applied at a constant rate of approximately 1 kgf/s. According to the manufacturer, the algometer has a margin of error of ±0.1 kg.

To ensure consistency in anatomical site selection, the cervical region was assessed with participants seated, at the midpoint between the spinous process of C7 and the lateral edge of the acromion. The thoracolumbar region was assessed in a prone position, at a point located 5 cm lateral to the spinous process of L1, over the erector spinae muscle. All measurements were performed on the same side, arbitrarily chosen, and repeated three times, with 30 s rest intervals between trials.

Subcutaneous adipose tissue thickness was measured in millimetres using a Baseline^®^ skinfold calliper (Fabrication Enterprises, White Plains, NY, USA), at the same anatomical sites (upper trapezius and lumbar region), following standardised procedures [26].

### 2.4. Procedures

Data collection took place in a controlled environment within the facilities of the research laboratory. Ambient conditions were kept stable, with temperature maintained between 22 and 24 °C and relative humidity between 40 and 60%, as monitored by a digital thermohygrometer. Noise and visual distractions were minimised, and all sessions were conducted under consistent artificial lighting. Participants were instructed to remain seated and relaxed for at least 10 min prior to assessment to allow for physiological acclimatisation. All procedures were carried out by the same researcher to minimise inter-rater variability [27].

Before starting the main assessments, anthropometric data, including body mass and height, were recorded for each participant. These values were used to calculate the Body Mass Index (BMI), expressed in kg/m^2^, using the standard formula BMI = weight (kg)/[height (m)]^2^ [28].

To ensure consistency and accuracy in the use of instruments and methodological procedures, a pilot study was conducted with six healthy volunteers. This preliminary phase enabled the standardisation of all procedures, including the use of the algometer (pressure application rate, positioning), the skinfold calliper (grasping technique, timing, and application angle), the ultrasound scanner (probe adaptation and imaging parameters), and the structured questionnaire used to assess sociodemographic data, general health status, and lifestyle habits. Although this questionnaire was not a validated instrument, it was developed by the research team and tested in the pilot phase to ensure clarity, feasibility, and relevance to the target population.

Intra-observer reliability was assessed through repeated measurements of pressure pain threshold and skinfold thickness, taken three days apart by the same examiner. The results showed excellent reliability, with ICC values of 0.96 and 0.98, respectively.

Data collection began with the assessment of PPT, mechanical pressure, and adipose tissue thickness, conducted in a randomised order to minimise measurement bias. All measurements were performed by a single trained examiner using a consistent protocol and imaging orientation, ensuring high methodological reproducibility.

Mechanosensitivity was assessed by evaluating PPT at two anatomical locations: the upper trapezius and the lumbar paraspinal region, bilaterally [29]. Pressure was applied perpendicularly to the skin at a constant rate of approximately 1 kgf/s until the participant reported the first sensation of pain. The mean of three consecutive measurements at each site was recorded, with a 30 s rest interval between repetitions [30]. To ensure standardisation across participants, specific anatomical landmarks and participant positions were used for each measurement:Cervical region: Participants were seated on a chair, with their feet flat on the floor and hands resting on their thighs [30]. Given that the deep fascia of the cervical region involves the upper trapezius muscle [31], the measurement site was defined as the midpoint between the spinous process of the C7 vertebra and the lateral edge of the acromion [30].Thoracolumbar region: Participants were placed in a relaxed prone position. The measurement site corresponded to the erector spinae muscle, 5 cm lateral to the spinous process of the L1 vertebra [32].

While ideally both regions would be assessed in the same position, this was not feasible due to anatomical constraints and differences in muscular orientation and accessibility between the cervical and lumbar regions. Supine positioning does not provide reliable acoustic windows or palpation access for either of the targeted fascia. Therefore, the seated and prone positions were selected based on established protocols in the literature and were validated during our pilot study in terms of reproducibility and image quality.

The order of assessments (cervical or lumbar first) was randomised across participants to minimise potential order effects or fatigue.

Regarding side selection, ultrasound-assisted fascial compression was performed unilaterally, with the side randomised for each participant. This decision was based on prior literature reporting no significant differences in fascial thickness or mechanosensitivity between dominant and non-dominant sides in healthy individuals [33]. Additionally, the technical complexity and time constraints of simultaneous ultrasound and algometry limited the feasibility of bilateral assessment.

The algometer was then combined with the ultrasound probe to evaluate the mechanical pressure required to reach the deep cervical and thoracolumbar fasciae. These non-diagnostic procedures were performed by a radiologist with 25 years of experience.

The power and overall gain of the ultrasound machine were adjusted to optimise the visualisation of the fascia and muscles and to obtain the best possible visualisation. The probe was placed on the skin to avoid tissue compression but also stable enough to maintain adequate contact between the probe and skin for consistent images. Pressure was applied perpendicularly and progressively increased until fascial contact was confirmed through real-time ultrasound imaging in the longitudinal plane (Figure 2 and Figure 3). The value displayed at that precise moment was recorded. These measurements were repeated three times, with 30 s intervals between each.

Adipose tissue thickness was measured by manually grasping a skinfold using the thumb and index finger of the left hand, while applying the adipometer perpendicularly with the right hand to the axis of the fold. The fold was maintained for approximately two seconds, after which the value shown on the adipometer was recorded.

Three consecutive measures were taken, with a 15 s interval between them to allow the skin and underlying tissues to return to their baseline state and minimise measurement bias due to tissue compression. This interval is consistent with procedures used in similar anthropometric studies. The order of skinfold assessments was randomised across anatomical sites to prevent systematic bias or tissue fatigue effects. The mean of the three measurements was used in subsequent analyses of both pressure pain threshold and adipose tissue thickness [34].

### 2.5. Statistical Analysis

For each outcome variable—PPT, adipose tissue thickness, and mechanical pressure required to reach the deep fasciae—the mean value of three consecutive measurements was calculated. Before proceeding with statistical analysis, the dataset was checked for completeness, input errors, and distribution characteristics. Regarding the mechanical pressure variable, considering that the ultrasound transducer had a surface area of 5 cm^2^ and the algometer used to assess PPT had a surface area of 1 cm^2^, the mechanical pressure values were normalised by area. Accordingly, the mechanical pressure values were divided by 5, resulting in the study variable ‘mechanical pressure’.

Data analysis was performed using IBM SPSS Statistics version 30.0 (IBM Corporation, Armonk, NY, USA), with the significance level set at 0.05. The normality of continuous variables was assessed using the Shapiro–Wilk test. Descriptive statistics were used to summarise the data. Means and standard deviations were reported for normally distributed variables, while medians and interquartile ranges (25th and 75th percentiles) were calculated when the assumption of normality was not met.

To compare the mechanical pressure required to reach the deep fascia between the cervical and lumbar regions, either a paired-samples *t*-test or the Wilcoxon signed-rank test was applied, depending on the distribution of the data.

To examine associations between PPT, mechanical pressure and adipose tissue thickness, correlation analyses were conducted using Pearson’s or Spearman’s correlation coefficients, according to data normality. These analyses were performed separately for the cervical and lumbar regions.

Additionally, simple linear regression analyses were performed to evaluate whether PPT or adipose tissue thickness individually predicted the mechanical pressure required to reach the deep fascia.

## 3. Results

The sample consisted of 43 participants, 23 females and 20 males, whose characteristics are summarised in Table 1.

The results of descriptive analysis of the mechanosensitivity (PPT), mechanical pressure, and adipose tissue thickness in both cervical and thoracolumbar fasciae are shown in Table 2.

Paired-samples analyses were conducted to compare values between the cervical and lumbar regions. While the measurements were performed under different postural conditions (seated for cervical; prone for lumbar), this comparison was maintained for descriptive purposes only and should be interpreted with caution. The values of mechanosensitivity were significantly higher in the cervical region (*p* < 0.001), and lumbar adipose tissue thickness was significantly greater than cervical thickness (*p* < 0.001; Wilcoxon signed-rank test).

Pearson’s correlation between the mechanical pressure required to reach the deep cervical fascia and the mechanosensitivity showed a weak, non-significant positive correlation (r = 0.262, *p* = 0.089) (Table 3). This finding was consistent with the untransformed data, suggesting a possible trend but no statistically significant association.

In the cervical region, Spearman’s rank correlation showed a very weak, non-significant association between adipose tissue thickness and the mechanical pressure (ρ = 0.134, *p* = 0.392). This suggests that subcutaneous fat in the cervical region may have little to no influence on mechanical pressure in this sample.

In the thoracolumbar region, Spearman’s rank correlation revealed a very weak and non-significant association between the mechanosensitivity and the pressure required to reach the deep fascia (ρ = 0.139, *p* = 0.375). This suggests that in this sample, mechanosensitivity was not meaningfully associated with the applied pressure to access the lumbar fascia.

Spearman’s correlation revealed a weak, non-significant negative association between lumbar adipose tissue thickness and mechanical pressure (ρ = −0.214, *p* = 0.169), suggesting that subcutaneous fat did not meaningfully influence the pressure to access the thoracolumbar fascia.

A simple linear regression was conducted to evaluate whether mechanosensitivity and adipose tissue thickness predict the mechanical pressure required to reach the deep cervical and thoracolumbar fasciae. Figure 4 and Figure 5 show the simple linear regression for the cervical and thoracolumbar regions, respectively.

Regarding mechanosensitivity in the cervical region, the model showed a non-significant trend toward a positive association (*p* = 0.089), explaining 6.9% of the variance (adjusted R^2^ = 0.046). Mechanosensitivity was not a statistically significant predictor (B = 0.053, *p* = 0.089).

Also, the model for the adipose tissue thickness was not statistically significant (*p* = 0.289) and explained only 2.7% of the variance (adjusted R^2^ = 0.004). Adipose tissue thickness did not significantly predict fascial pressure (B = 0.671, *p* = 0.289).

In the lumbar region, for the mechanosensitivity, the model was not statistically significant (*p* = 0.785) and explained only 0.2% of the variance (adjusted R^2^ = −0.023). Mechanosensitivity was not a significant predictor (B = 0.004, *p* = 0.785).

To examine whether lumbar adipose tissue thickness predicts the pressure required to reach the deep fascia, the model was not statistically significant (*p* = 0.257), explaining only 3.1% of the variance (adjusted R^2^ = 0.008).

Adipose tissue thickness did not significantly predict fascial pressure (B = −0.604, *p* = 0.257).

## 4. Discussion

This study investigated the mechanical pressure required to reach the deep cervical and thoracolumbar fasciae, examined its association with mechanosensitivity and adipose tissue thickness, and determined whether mechanosensitivity and adipose tissue thickness can predict the mechanical pressure needed to reach the fascia, using linear regression analysis.

### 4.1. Regional Patterns and Tissue Behaviour

While established anatomical principles suggest that both mechanosensitivity and superficial tissue thickness could influence fascial accessibility, our findings challenge these assumptions and offer new perspectives for clinical reasoning and future research.

Consistent with previous anatomical and imaging studies, the lumbar region demonstrated significantly higher adiposity and reduced mechanosensitivity than the cervical region [35]. Surprisingly, these regional differences did not yield corresponding variation in the pressure required to reach deep fascia, suggesting that fascial accessibility may depend less on superficial characteristics than one can assume. It is important to clarify that the cervical and lumbar fasciae were assessed under different postural conditions—seated and prone, respectively—which may have influenced local muscle tone and tissue behaviour. Therefore, direct comparisons between regions should be interpreted with caution. Nonetheless, we present these regional data side-by-side not as strict comparisons, but to illustrate that—despite differing superficial profiles—both regions required similar mechanical pressure to reach the deep fascia. This unexpected finding raises important questions about the assumed influence of overlying tissues and supports the need for further targeted investigation.

The results of our study demonstrated that the mechanical pressure required to reach the deep cervical and thoracolumbar fasciae was similar, approximately 6 N/cm^2^. Notably, participants exhibited mechanosensitivity values higher than the pressure required to engage fascia, indicating that fascial engagement may occur before pain threshold is exceeded, which was also found in the present study, and is a valuable consideration in pain-sensitive populations. This evidence is particularly relevant considering that some myofascial induction techniques are based on the application of sustained mechanical pressure on the fascia [36]. As the mechanical pressure was substantially lower than the mechanosensitivity thresholds observed in the cervical (18.88 ± 0.917 N/cm^2^) and lumbar (46.46 ± 2.408 N/cm^2^) regions, the myofascial technique may represent an effective therapeutic approach, potentially capable of reaching the fascia without inducing symptomatology. However, it is important to note that the participants were healthy and asymptomatic, which may limit the generalisability of the results to populations with pain or associated pathologies. Furthermore, the role of epidermal cells was not directly assessed in this study. These cells, innervated by autonomic Aδ and C fibres, may contribute to pain modulation and functional recovery through their involvement in neuroimmune and neurosensory mechanisms [11,12]. Therefore, their potential influence during fascial compression should not be overlooked.

When we compared mechanosensitivity between the cervical and thoracolumbar fasciae, we observed that the cervical region showed significantly greater mechanosensitivity, which could possibly be explained by various anatomical, neurophysiological and functional factors. One such factor is the involvement of the epidermis, which contains dense networks of free nerve endings—particularly Aδ and C fibres—associated with nociceptive and autonomic signalling. Although the epidermis was not the primary focus of this study, its stimulation is unavoidable during manual pressure application. Recent studies, including those by Selva-Sarzo et al. [21] and Talagas et al. [12], suggest that epidermal stimulation can modulate pain perception and enhance functional outcomes, underscoring its relevance when interpreting regional differences in mechanosensitivity. The introduction of this concept at this stage aims to integrate superficial neuroanatomical contributions to our findings, thereby complementing earlier discussions on tissue innervation.

Marzvanyan & Alhawaj [37] state that the type and density of innervation, specifically receptors, can vary according to the different regions of the body, with the cervical region having a higher concentration of sensory receptors and free nerve endings, which can make it more sensitive to mechanical stimuli. In addition, from a biomechanical point of view, the cervical region is more mobile, while the lumbar region is more robust and adapted to withstand greater loads, which may influence tissue sensitivity, making the cervical region more susceptible to lower intensity mechanical stimuli. Also, the thickness of the fascia and the thickness of the tissue can facilitate the transmission of mechanical stimuli to nociceptors, which in a way seems to agree with the results of our study, as the thickness of adipose tissue was significantly lower in the cervical region when compared to the lumbar region. On the other hand, postural factors may also be at the core of these differences, because the cervical region is more prone to chronic postural tensions, particularly in populations that spend a lot of time sitting or using electronic devices, as is the case with the participants in our sample. In this context, repetitive and/or sustained posture and physical exercise can increase the thickening of the fascia and the release of chemical substances that can sensitise the receptors, which may explain the greater mechanosensitivity in the cervical region, an idea corroborated by Martinez-Merinero et al. [38] who found that individuals with cervical anteriorisation showed increased mechanosensitivity. Some physical and chemical properties of the hyaluronic acid found between the deep fascia and the muscle may also be the basis of the differences found between the regions studied, as they may contribute to changes in the viscoelasticity, mechanical plasticity and non-linear elasticity of the fascia’s extracellular matrix. Thus, the biomechanical properties of connective tissue can change due to the presence of lactic acid resulting from exercise, as well as the decrease in pH, which can favour thickening and alter the sensitivity of receptors [39,40]. As for the thickness of the adipose tissue, the fact that the neck is not a typical region for fat accumulation may have justified significantly lower values when compared to the lumbar region, which is more likely to accumulate fat. However, the results showed no statistically significant association with mechanical pressure, which suggests that the pressure that is transmitted is not absorbed by the fat, allowing pressure to be transferred to deeper tissues.

### 4.2. Mechanosensitivity, Adipose Tissue Thickness and Predictive Value

No significant correlations were observed between mechanosensitivity or adipose tissue thickness and fascial pressure in either study. This contradicts the expectation that thicker adipose layers act as a mechanical buffer. As Chaudhry et al. [41] noted, pressure is not “absorbed” by fat tissue in a way that impedes fascial deformation; adipose tissue is highly deformable and allows transmission of pressure to deeper layers. Recent biomechanical models and experimental compression tests have demonstrated that subcutaneous adipose tissue exhibits low resistance and high deformability under vertical load, allowing for efficient force transmission into deeper structures such as fasciae. This supports the interpretation that adipose tissue does not impede fascial deformation and reinforces its minor role as a mechanical buffer [42].

Interestingly, the lack of significant correlations between mechanosensitivity, adipose tissue thickness, and the mechanical pressure required to access the fascia is a particularly relevant finding. It challenges the common clinical assumption that superficial adiposity predicts the force needed for effective manual intervention. Clinicians often adjust pressure based on palpated fat thickness or perceived tenderness, yet our results suggest that these variables may not reliably indicate fascial accessibility [43]. This discrepancy calls for a reassessment of such assumptions and highlights the need for more objective criteria to guide pressure modulation during manual therapies. Theoretical explanations for this lack of association may include the viscoelastic behaviour of connective tissues, inter-individual variability in fascial architecture, and the role of other less visible variables such as hydration, hyaluronic acid concentration, or even neural modulation [44]. Recognising these complexities can lead to more nuanced and evidence-informed manual therapy practices.

The absence of correlation with mechanosensibility may also reflect the influence of participant characteristics. A large proportion of the sample engaged in regular physical activity, which could have contributed to elevated pain tolerance and altered tissue sensitivity. The literature shows that individuals with greater muscle strength tend to exhibit higher pain thresholds, a factor that might explain the lack of association between PPT and fascial access pressure [45]. Additionally, the posture adopted during assessment may have influenced mechanosensitivity measurements. While the lumbar region was assessed in a relaxed prone position, the cervical region was evaluated in a seated posture, potentially involving greater muscle activation. This difference in physiological context may have attenuated the comparability between regions and could partly explain the lack of association between pressure pain threshold and fascial access pressure. Future studies may benefit from standardising participant positioning—such as using a supine posture—to minimise muscular interference and improve methodological consistency.

Additionally, muscle mass and physical activity levels can influence subcutaneous fat distribution. Participants with higher muscle tone may exhibit lower adiposity and increased fascia compliance [46], further hampering the ability to establish a relationship between superficial and deep tissues. These insights reinforce the idea that surface-level measures alone are insufficient to predict mechanical outcomes, especially in healthy or athletic populations such as the one in the present study.

### 4.3. Considerations and Future Directions

The findings of our study carry important clinical implications. The ability to reach deep fascia before triggering nociceptive responses suggests that fascial techniques can be applied effectively and safely. Moreover, the lack of a strong influence from superficial adiposity implies that manual therapy protocols may be broadly applicable without substantial individual adjustment, aligning with recent clinical guidance for standardised practice [47].

These insights expand upon the neurophysiological models of fascia, which recognise its role in proprioception, interoception, and nociception [48]. Given that fascia deformation occurs prior to discomfort, manual pressure may produce local and systemic effects without provoking pain, contributing to functional improvements in patients with back pain or myofascial dysfunction.

However, this study has some limitations. First, the influence of cutaneous structures—particularly the epidermis—was not directly assessed. Reflexes mediated by epidermal cells, which are innervated by Aδ and C fibres and known to modulate pain and mobility, may have contributed to the observed responses. Second, although care was taken to standardise procedures, slight variations in probe positioning are inevitable during manual application. These small angular deviations or shifts may compromise the reproducibility of pressure readings. Finally, the lack of direct control over the stimuli applied to superficial tissues further limits the interpretation of the mechanosensory response as being fascia-specific.

Moreover, the relatively small sample size and the inclusion of only healthy young adults restrict the generalisability of our findings. The inclusion of symptomatic individuals—particularly those with chronic musculoskeletal conditions—could offer greater insight into the clinical relevance of fascial mechanosensitivity. Future research should explore these mechanisms in clinical populations and assess potential interactions between fascial architecture and neuromuscular factors.

Although this study examined the cervical and thoracolumbar regions, it is important to acknowledge that a direct comparison between them is not methodologically valid due to differences in participant positioning and muscular activation during assessment. Nonetheless, both regions were included to characterise regional variability in fascial accessibility and to provide foundational data for future research. The inclusion of dynamic assessments and shear-wave ultrasound elastography could offer more refined insights into fascial behaviour under varying mechanical loads. Future investigations should also explore additional anatomical regions to determine whether the observed patterns are consistent across different body areas, thereby providing a more comprehensive understanding of fascial engagement in clinical and non-clinical populations. Moreover, given the cross-sectional nature of this study, longitudinal designs are warranted to examine how mechanosensitivity, adiposity, and fascial behaviour evolve over time, particularly in response to pain or rehabilitation interventions. This would enhance our understanding of the temporal dynamics underlying fascia-related dysfunction and recovery.

Due to laboratory constraints and time limitations during data collection, it was not feasible to record additional participant characteristics such as handedness, limb dominance, or physical activity habits (e.g., frequency, intensity, or whether the activity was predominantly unilateral or bilateral). Although the side of assessment was randomised or arbitrarily chosen depending on the procedure, these individual characteristics may influence baseline muscle tone and fascial properties, particularly in regions subjected to habitual loading or asymmetrical use. The absence of this information limits the ability to fully control for or interpret potential sources of variability. Future studies should consider including these variables to enhance the understanding of individual differences in mechanosensitivity and tissue thickness.

In clinical terms, the present findings encourage a more standardised and evidence-based application of fascial techniques, especially in settings where manual pressure is used therapeutically. The fact that deep fascia can be engaged before pain thresholds are exceeded reinforces the safety of these techniques, even in patients with heightened pain sensitivity. Moreover, the absence of a significant role of adipose thickness or mechanosensitivity in predicting fascial access pressure suggests that clinicians should be cautious about relying on superficial palpation or visual estimates when modulating force. Instead, integrating objective tools or structured protocols may enhance treatment efficacy and reduce variability between practitioners. These insights are particularly relevant for manual therapists, physiotherapists, and osteopaths aiming to optimise outcomes while minimising discomfort and tissue overload.

## 5. Conclusions

This study sets out the mechanical pressure required to reach the deep cervical and thoracolumbar fasciae and to evaluate whether subcutaneous adipose tissue thickness and pressure pain threshold could predict this pressure. 

The mechanical pressure required to reach the deep cervical and thoracolumbar fasciae was similar, with an average value of approximately 6 N/cm^2^, suggesting a consistent mechanical response across these anatomical regions. Mechanosensitivity was higher in the deep cervical fascia, whereas subcutaneous adipose tissue thickness was greater in the thoracolumbar region; however, these differences do not significantly influence fascial accessibility.

This study supports a more inclusive and standardised application of manual therapies, reducing reliance on body composition as a guide for treatment decisions.

Ultimately, by demonstrating that commonly assumed predictors such as adipose thickness and pressure pain threshold do not significantly influence fascial access, this research invites a re-evaluation of clinical reasoning surrounding soft tissue therapy. More importantly, this study positions fascia not merely as a passive structural element, but as a dynamic, context-sensitive interface. Its behaviour may be modulated by superficial layers, such as the epidermis, through reflexive and autonomic pathways. These mechanisms can influence mechanosensitivity, contribute to mobility limitations, and intensify pain perception. Therefore, integrated assessments that consider both structural and neurophysiological dimensions are warranted to better inform rehabilitation strategies.

## Figures and Tables

**Figure 1 sensors-25-05073-f001:**
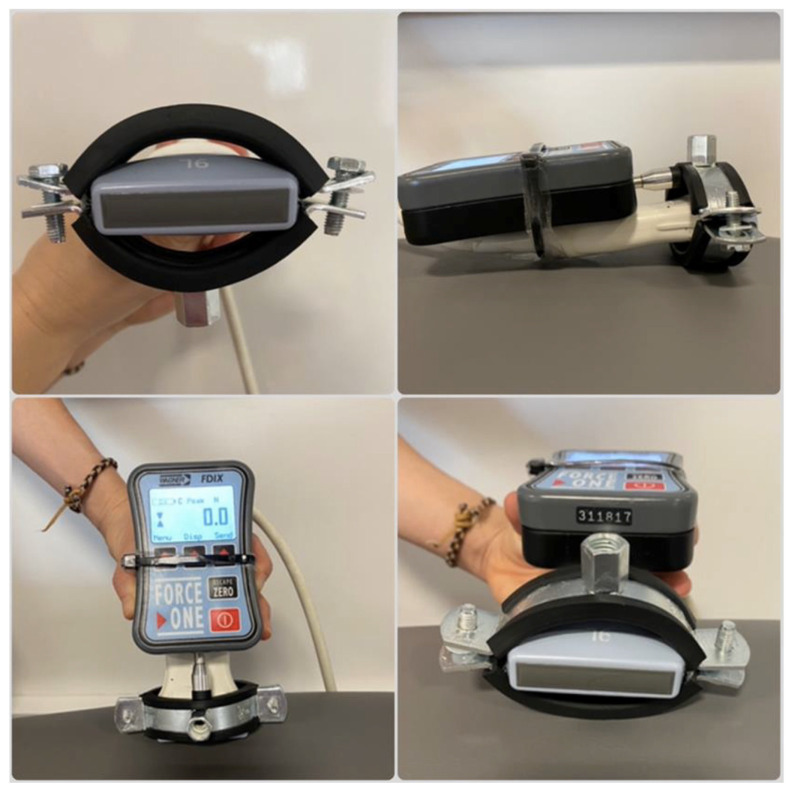
Custom-designed device used to couple the ultrasound probe with the algometer, allowing simultaneous measurement of applied mechanical pressure and real-time visualisation of fascial deformation.

**Figure 2 sensors-25-05073-f002:**
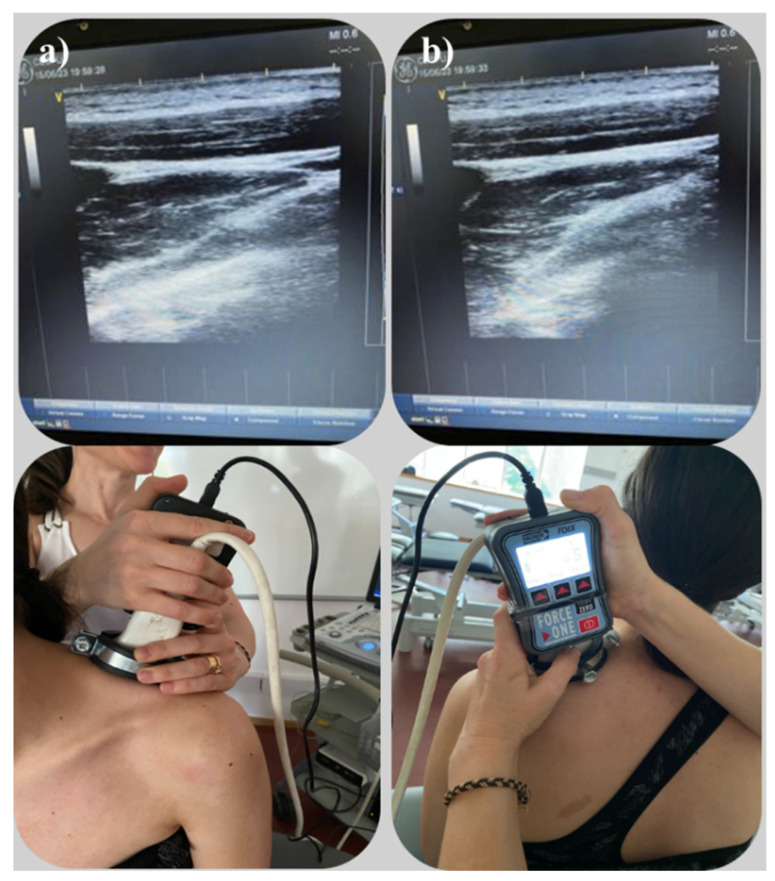
Ultrasound image and probe coupled with the algometer in the cervical region: (**a**) before pressure application and (**b**) after the application of pressure resulting in deformation of the deep cervical fascia. These images illustrate the methodological procedure used to identify fascial deformation in real time, validating the point at which the recorded mechanical pressure corresponds to fascial engagement.

**Figure 3 sensors-25-05073-f003:**
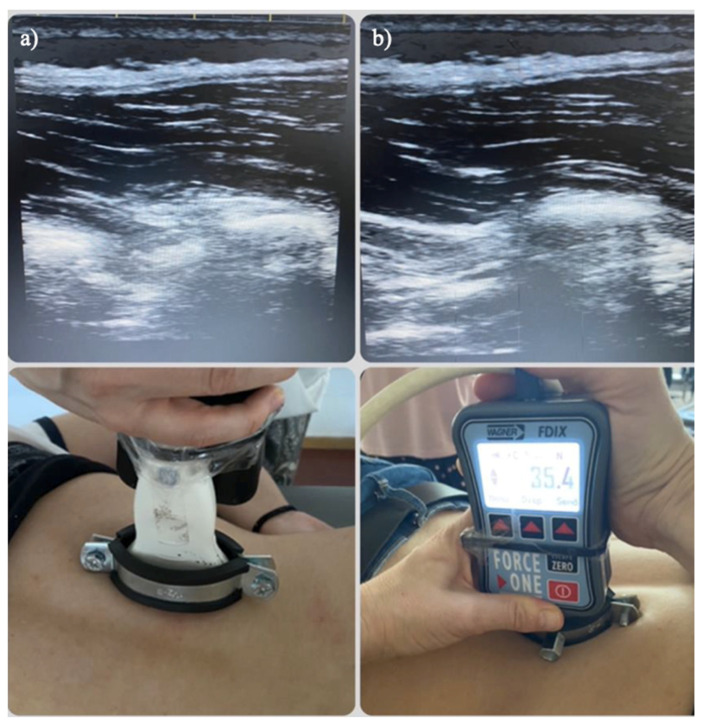
Ultrasound image and probe coupled with the algometer in the thoracolumbar region: (**a**) before pressure application and (**b**) after the application of pressure resulting in deformation of the deep thoracolumbar fascia. These images demonstrate the consistency of the procedure across different anatomical regions, supporting the comparison of fascial accessibility in areas with distinct structural characteristics.

**Figure 4 sensors-25-05073-f004:**
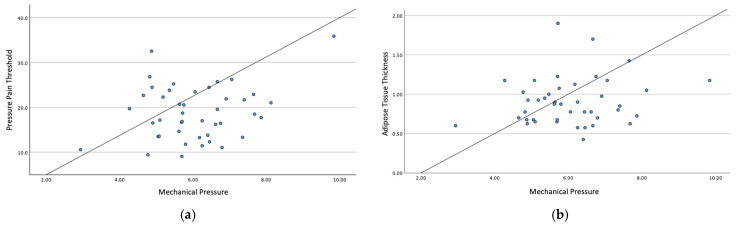
(**a**) Simple linear regression analysis between PPT and mechanical pressure for the cervical region; (**b**) simple linear regression analysis between adipose tissue thickness and mechanical pressure for the cervical region.

**Figure 5 sensors-25-05073-f005:**
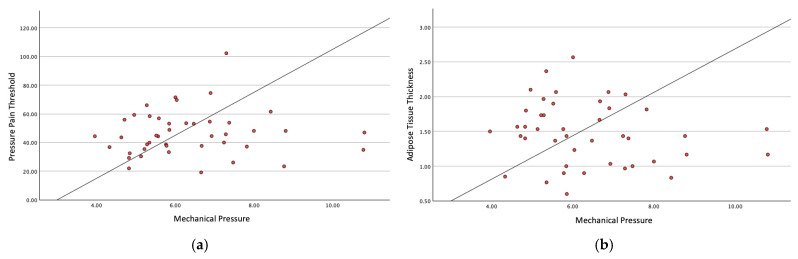
(**a**) Simple linear regression analysis between PPT and mechanical pressure for the thoracolumbar region; (**b**) simple linear regression analysis between adipose tissue thickness and mechanical pressure for the thoracolumbar region.

**Table 1 sensors-25-05073-t001:** Participants’ characteristics: mean and standard deviation values for height, weight and IMC, and median and 25th and 75th percentiles for age.

Variable	Mean (SD)	Med (P25; P75)
Age (years)		21 (19; 24)
Weight (Kg)	64.77 (12.793)	
Height (m)	1.70 (0.095)	
BMI (Kg/m^2^)	22.38 (3.565)	

Med—median; P25—25th percentile; P75—75th percentile; Kg—kilogram; m—metre; Kg/m^2^—kilogram per square metre.

**Table 2 sensors-25-05073-t002:** Mean and standard deviation values, and median and 25th and 75th percentiles of mechanical pressure, pressure pain threshold and adipose tissue thickness in both cervical and thoracolumbar fasciae.

Variable	Mean (SD)		Med (P25; P75)		*p*-Value
	Cervical	Thoracolumbar	Cervical	Thoracolumbar	0.80 *
Mechanical pressure (N/cm^2^)	6.06 (0.186)			5.85 (5.280; 7.287)	<0.001 #
Pressure pain threshold (N/cm^2^)	18.88 (0.917)	46.46 (2.408)			
Adipose tissue thickness (cm)		1.48 (0.069)	0.88 (0.675;1.08)		<0.001 *

Med—median; P25—25th percentile; P75—75th percentile; N/cm^2^—Newton per square centimetre; cm—centimetre; * Wilcoxon test; # Student *t*-test.

**Table 3 sensors-25-05073-t003:** Pearson and Spearman’s correlation analysis of mechanical pressure and pressure pain threshold, and mechanical pressure and adipose tissue thickness for cervical and thoracolumbar regions.

Region		r/ρ; *p*
Cervical	Mechanical Pressure vs. PPT	0.262 *; 0.089
	Mechanical Pressure vs. Adipose Tissue Thickness	0.134 #; 0.392
Thoracolumbar	Mechanical Pressure vs. PPT	0.139 #; 0.375
	Mechanical Pressure vs. Adipose Tissue Thickness	−0.214 #; 0.169

* r value for the Pearson correlation coefficient; # ρ value for Spearman’s correlation coefficient. PPT—pressure pain threshold.

## Data Availability

Data are unavailable due to privacy or ethical restrictions.

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
