# Peer review of "Mechanosensitivity and Adipose Thickness as Determinants of Pressure to Reach Deep Fasciae in Cervical and Thoracolumbar Regions"

_sensors, 2025, doi:10.3390/s25165073_

Round 1
Reviewer 1 Report
Comments and Suggestions for Authors
I would like to thank MDPI for considering me as a reviewer for this manuscript. It is an interesting study aimed at contributing to the prediction of the mechanical pressure required to access the targeted fascia using validated measurement tools.
The topic addressed requires scientific clarification, and the presented information appears relevant to fascial therapy, potentially helping to reduce some of the current knowledge gaps in the field. This is especially pertinent given the low reproducibility of many previously published studies and the frequent oversight of other systems or receptors also stimulated during manual therapy, which can influence fascial response. The conclusions of this manuscript are likely to be of interest to readers and healthcare professionals who apply fascial techniques in clinical practice.
However, the manuscript requires several improvements, particularly in clarifying methodological concepts, explaining the statistical procedures, and restructuring the text to ensure logical coherence and clarity. These aspects are detailed below.
I consider that the current study design does not allow for a joint hypothesis test comparing the cervical and lumbar regions, as the assessments were conducted under differing muscle tone conditions: one in a prone position and the other in a sitting position. Therefore, a direct comparison between regions is not methodologically appropriate.
Specific Comments:
Introduction
-
Line 39: It is necessary to introduce and cite that the fascia has both innervation and vascularization.
-
Line 57: Nociceptive input originates from the epidermis, dermis, and deeper tissues. Please elaborate on this with proper references. I suggest including the work of Talagas on the epidermis.
-
Lines 62–66: While fascia and subcutaneous tissue are discussed, the skin is not mentioned. It is essential to include a discussion of cutaneous structures, as any manual pressure applied to access the fascia must first pass through the skin layers—epidermis, dermis, subcutaneous tissue—before reaching the fascia. A description of their innervation is also necessary. The epidermis, due to its autonomic innervation (Aδ and C fibers), and the dermis, via somatosensory Aβ fibers, play distinct roles.
It is important to emphasize that during manual therapy, only the epidermis is in direct contact with the practitioner's hand, although deeper structures are indirectly influenced. It would be beneficial to include an embryological rationale: the epidermis derives from the ectoderm, like the central nervous system, and is innervated by the autonomic nervous system (via free nerve endings). In contrast, the dermis originates from the mesoderm and is innervated by the peripheral somatic nervous system.
Materials and Methods
2.1. Study Design and Ethics
-
Line 88: Please clarify whether the single intervention session was conducted at the same time of day for all participants or if it was randomized, as circadian influences could alter fascial tone.
2.2. Participants
-
Line 101: The rationale for the pragmatic sampling approach should be clearly and convincingly justified.
2.3. Instruments
-
Lines 107–108: It is necessary to specify the validated questionnaire used to collect data on participants’ lifestyle and general health status.
-
Lines 115–117: Considering individual differences in hydration status and past dysfunctions, the pressure needed to reach the same fascial depth may vary significantly. Please explain how reproducibility was ensured when applying pressure using the pressure pain threshold (PPT) device while observing ultrasound images. What is the margin of error for this procedure?
-
Lines 117–119: How is the loss of fascial glide or fascial plane deformation defined and measured reproducibly?
Additionally, how is consistency ensured in terms of the anatomical site being measured across all participants?
2.4. Procedures
-
Line 143: The term “controlled environment” needs to be defined. Which variables were controlled? It is important to specify environmental parameters such as ambient temperature, humidity, and the subject’s acclimatization period before assessment.
-
Line 147: Please indicate the validated formula used to calculate body mass index (BMI).
-
Lines 148–153: The procedures must be described in a precise and reproducible manner.
-
Lines 156–158: Specify the validated questionnaire employed in the assessment.
-
Line 162: Explain the standardization methods used to ensure reproducibility of the measurements across participants.
-
Line 164: The procedures must include the participant's position during measurement, the exact anatomical location of the measurement point, and the anatomical landmarks used for identification.
-
Lines 163–178: The order and execution of measurements in both the lumbar and cervical regions must be clarified. In the lumbar region, measurements were performed in the prone position, which implies minimal muscle activation. In contrast, cervical region measurements were conducted in a seated position, where muscle activation is present. Please explain this discrepancy. If the objective is to assess the deep fascia, why were both regions not assessed in the same posture, for example, in the supine position? Since the two anatomical regions were evaluated under different physiological conditions, their comparison lacks validity.
Which region was assessed first—the cervical or lumbar? Provide justification for this decision. -
Line 178: The choice of side for assessment must be explained and referenced. Why was a unilateral assessment conducted instead of a bilateral one?
-
Line 188: This information appears duplicated in this section: “These measurements were repeated three times, with 30-second intervals between each.” Please remove the repetition.
-
Line 207: Justify and reference the reason for selecting a 15-second interval between measurements.
Please also clarify whether the sequence of measurements was fixed or randomized, and provide the rationale behind this decision.
Results
-
Lines 254–257: Given that the two regions were assessed under different conditions, the comparison lacks methodological validity. Please justify the rationale for making this comparison despite the differing assessment conditions.
-
Line 278: Consider whether the absence of significant differences might be attributed to the differing postures during assessment. Sitting posture involves greater muscular activity and tone compared to the prone position. Therefore, lumbar measurements taken at rest are not directly comparable to cervical measurements obtained during muscular activation.
Discussion
-
Lines 332–336: Reiterate that a valid comparison between both regions is not feasible due to differences in muscle activation. Clarify the justification for maintaining the comparison, or consider removing it.
-
Line 350: Please add that epidermal cells, innervated by the autonomic nervous system via Aδ and C fibers, were not taken into account in this study. Their involvement could influence the results, as stimulation of these cells has been shown to reduce pain and improve mobility. Relevant references can be provided if needed.
-
Line 357: This line likely refers to free nerve endings, which are located in the epidermis. Include a rationale for the introduction of the epidermis at this stage, despite its absence in earlier sections. Consult Talagas’ work to support this explanation and provide coherence.
-
Lines 396–398: These observations may support the argument for conducting measurements in the supine position to avoid muscle activation.
-
Line 418: Add to the study’s limitations the lack of control over stimuli applied to the epidermis, as their potential effects were not considered. Moreover, reflexes mediated by epidermal cells could have influenced the outcomes. The probe’s position during measurements was not fixed, which compromises reproducibility, as slight tilts or shifts are inevitable during manual assessments.
-
Line 427: Clarify that, based on the methodological design, a valid comparison between both regions is not feasible as currently described.
Author Response
Our Comments to Rev 1 are included in the uploaded file.

Reviewer 2 Report
Comments and Suggestions for Authors
This study investigates the mechanical pressure required to access the deep cervical and thoracolumbar fasciae, examining its relationship with mechanosensitivity and adipose tissue thickness. The authors present a well-structured and methodologically sound approach, utilizing a combination of an algometer, ultrasound scanner, and skinfold caliper to measure relevant variables. The results show interesting findings, particularly that superficial adiposity and mechanosensitivity do not significantly predict the pressure needed to engage the deep fascia, challenging common assumptions in the field.
While the study provides excellent foundational data, a few areas require further clarification.
- The study primarily uses healthy, young adult participants from a university setting. While this is valuable for controlling various variables, the generalizability of the results to clinical populations, especially those with chronic pain or musculoskeletal conditions, is limited. Future studies should include symptomatic individuals to better understand the application of the results in clinical settings.
- The absence of significant correlations between mechanosensitivity, adipose tissue thickness, and mechanical pressure required to access the fascia is an important finding but requires deeper interpretation. Given that clinical practices often rely on adiposity as a predictor for manual therapy techniques, the lack of correlation challenges conventional assumptions. This finding could be better contextualized, and further explanations or theoretical implications should be discussed to enhance the clinical relevance of the study.
- The study excludes individuals with conditions like fibromyalgia or chronic neck and lumbar pain. The role of these conditions in influencing mechanosensitivity or adipose thickness and their effect on fascia accessibility should be further explored. Including such populations might provide more clinically relevant data.
- The study focuses on two anatomical regions (cervical and lumbar), while other regions of the body could potentially show different relationships between mechanosensitivity, adipose thickness, and fascia pressure. Including additional anatomical regions could provide a broader perspective on fascial engagement across the body.
- The cross-sectional design provides a snapshot but does not account for any potential changes over time. Longitudinal studies could provide more insights into the temporal relationship between mechanosensitivity, adipose tissue thickness, and fascia accessibility, especially in relation to pain management or therapeutic interventions.
- Some of the figure legends (e.g., Figures 2 and 3) could benefit from more detailed descriptions. Currently, they briefly mention the deformation of the fascia but do not explain the relevance of these images to the study findings or how they directly connect to the study's results.
- In the "Materials and Methods" section, the description of the combination of the ultrasound and algometer could be more detailed. While the method is introduced, it could benefit from clearer explanations of the exact measurement process, such as how the ultrasound’s surface area influenced the mechanical pressure readings.
- The paper mentions that adipose tissue is highly deformable and allows pressure to be transmitted to deeper tissues, which is a significant point. However, more references or empirical evidence to support this assertion would strengthen the argument. More studies specifically targeting the influence of adipose tissue on fascial pressure would enhance the credibility of these conclusions.
- The simple linear regression models in the results section show weak associations between the predictors and mechanical pressure. It would be helpful to discuss the implications of these findings more thoroughly, particularly why these weak correlations do not imply a relationship despite theoretical assumptions.
- The clinical implications of this study's findings could be expanded further.
- Including a more detailed section on future research directions would be beneficial. Specifically, exploring dynamic pressure variations in clinical populations, examining the role of muscle tone in fascial pressure dynamics, and investigating the effects of physical activity on mechanosensitivity could provide more comprehensive insights into the topic.
Author Response
Our Comments to Rev 2 are included in the uploaded file.

Round 2
Reviewer 1 Report
Comments and Suggestions for Authors
I appreciate and commend the authors for the considerable effort made in revising their manuscript, especially considering the complexity of accurately updating some sections. I have modified certain paragraphs to better reflect current scientific understanding.
Below, I include the revised versions and suggest additional updates for consideration.
Lines 60–62:
The dermis is innervated exclusively by Aβ fibers and not by Aδ fibers. The paragraph may be more accurate if reformulated as follows:
"Nociceptive input does not arise exclusively from the fascia but from multiple tissue layers, including the epidermis, dermis, subcutaneous tissue, and deeper structures. These layers differ in embryological origin and innervation: the epidermis, derived from the ectoderm (like the central nervous system), is innervated by autonomic Aδ and C fibres responsible for nociception and thermal stimuli. In contrast, the dermis and subcutaneous tissues, of mesodermal origin, are innervated by somatic Aβ fibres that respond to mechanical stimuli. This layered neuroanatomical arrangement explains the varied sensory responses elicited by manual pressure and highlights that any attempt to reach the deep fascia inevitably stimulates multiple nociceptive sources. As such, isolating fascial contribution in vivo remains methodologically challenging [11,12]."
Lines 74–80:
The following revision is recommended:
"However, it is important to note that any manual pressure applied to reach the fascia must first pass through the overlying skin layers: the epidermis, dermis, and subcutaneous tissue. These structures are not merely passive barriers but actively contribute to the autonomic response—initially via the epidermis and subsequently through mechanosensory input from the dermis. During myofascial therapy, the practitioner's hand is in direct contact with the epidermis. While deeper structures such as the dermis, subcutaneous tissue, and fascia are the intended therapeutic targets, they are indirectly affected by the mechanical transmission of pressure [12]."
Lines 91–95:
Since you accurately mention pain and mobility, it would be appropriate to differentiate the role of the ectoderm-derived epidermis. Please consider integrating the following reformulated paragraph:
"However, studies are lacking that isolate the central autonomic effects of the epidermis on pain and mobility (https://doi.org/10.3389/fpain.2025.1525964), and subsequently correlate the mechanical pressure required to reach deeper fascial layers with tissue thickness and subjective pain perception. This knowledge could facilitate the development of evidence-based guidelines for more effective manual therapy interventions for neck and low back pain [8,21,22]."
Line 134–135:
Please indicate that validated questionnaires were not used to collect data. This should be explicitly acknowledged in the manuscript.
Lines 160–162 and 238–241:
It is necessary to specify whether the assessments were performed on the right or left side of the body. It would also be relevant to state whether participants were right- or left-handed to better understand potential baseline muscle tone differences in the muscular chains involved in pressure zones. Additionally, indicate whether participants engaged in physical activity and whether it was predominantly unilateral or bilateral. If this information was not recorded, these limitations should be clearly stated in the manuscript.
Line 440:
I think it is interesting to add the following reference next to the Talagas reference:
https://doi.org/10.3389/fpain.2025.1525964
Lines 558–559: It would be interesting to expand on the information proposed in the conclusions:
"More importantly, it positions fascia not merely as a passive structural element, but as a dynamic, context-sensitive interface that requires an integrated assessment, taking into account epidermal-generated reflexes that can affect the fascia, leading to mobility limitations and pain."
Author Response
The responses to Rev 2 Comments can be found in the .pdf file.
